# ADAPTIVE GRADUATED NON-CONVEXITY FOR POINT CLOUD REGISTRATION

## ABSTRACT

Point cloud registration is a critical and challenging task in computer vision. It is difficult to avoid poor local minima since the cost function is significantly non-convex. Correspondences tainted by significant or unknown outliers may cause the probability of finding a close-to-true transformation to drop rapidly, leading to point cloud registration failure. Many registration methods avoid local minima by updating the scale parameter of the cost function using graduated non-convexity (GNC). However, the update is usually performed in a fixed manner, resulting in limited accuracy and robustness of registration, and failure to reliably converge to the global minimum. Therefore, we present a novel method to robust point cloud registration based on Adaptive Graduated Non-Convexity (AGNC). By monitoring the positive definiteness of the Hessian of the cost function, the scale in graduated non-convexity is adaptively reduced without the need for a fixed optimization schedule. In addition, a multi-task knowledge sharing mechanism is used to achieve collaborative optimization of non-convex cost functions at different levels to further improve the success rate of point cloud registration under challenging high outlier conditions. Experimental results on simulated and real point cloud registration datasets show that AGNC far outperforms state-of-the-art methods in terms of robustness and accuracy, and can obtain promising registration results even in the case of extreme 99% outlier rates. To the best of our knowledge, this is the first study that explores point cloud registration considering adaptive graduated non-convexity.

## 1 INTRODUCTION

Point cloud registration is a critical and challenging task in computer vision. Its goal is to transform point clouds with arbitrary coordinate systems into a common coordinate system to obtain full coverage of an object or scene. Point cloud registration can be used for scene reconstruction (Yu et al., 2023; Mei et al., 2023), object recognition (Jiang et al., 2023; Yuan et al., 2024; Nie et al., 2024), autonomous driving (Lu et al., 2019; Liu et al., 2024), and medical imaging processing (Chen et al., 2022c; Ginzburg & Raviv, 2022; Ma et al., 2023).

The point cloud registration problem can be easily solved when the true correspondences between point clouds are known. But in reality, solvers yield subpar estimates since the correspondences are either uncertain or include a large number of outliers (Bustos & Chin, 2017; Chen et al., 2022b; Jiang et al., 2023). High outlier rates (sometimes exceeding 99%) are a typical feature of point cloud keypoint detection and registration, which poses a great challenge to point cloud registration (Huang et al., 2020; Qin et al., 2022; Yuan et al., 2023). This challenge is common, where matching often produces false correspondences due to noise, occlusions, and sensor errors. For example, in autonomous driving, LiDAR scanning is often interfered by dynamic objects such as cars and pedestrians and contain a lot of background noise (Bogdoll et al., 2022). Registration methods must effectively handle these outliers to ensure proper functioning of safety systems. Given the inherent ambiguity in point cloud data association and the potential measurement errors that may produce outliers, the performance of point cloud registration depends on how well it handles these outliers.

Over the past few decades, a lot of research has been done on point cloud registration with correspondences tainted by outliers. Typical methods are iteratively reweighted least squares (IRLS) (Wang et al., 2023; Huang et al., 2024), random sample consensus (RANSAC) (Fischler & Bolles, 1981; Barath & Matas, 2021), and M-estimators (Le & Zach, 2020; Li et al., 2023; Sidhartha et al., 2023). When the percentage of outliers in the input is low, a set of optimal parameters can be easily

obtained by minimizing the residual sum of squares, and the cost can be optimized using popular IRLS solvers. However, in the presence of a large number of outliers, standard IRLS with a fixed threshold often produces results that are biased toward the outliers. As a result, the transformation estimates are far from the ground truth transformation.

For outliers, RANSAC has been widely used for registration problems. The main reasons are its algorithmic simplicity and its ability to handle contaminated data containing more than 50% outliers. But there are still a few issues that need to be fixed. On the one hand, the random sampling has a slow convergence speed. On the other hand, the predefined inlier threshold leads to low accuracy in registering high proportion outlier point clouds. To address these issues, many variants have been proposed to speed up the computation time (Yang et al., 2021; Chen et al., 2022b), improve the solution stability (Zhang et al., 2023), and automatically determine the threshold (Wei et al., 2023).

M-estimators and IRLS are mathematically equivalent (He et al., 2013), and M-estimators are also sensitive to the threshold. However, the threshold can be determined heuristically based on the problem. One approach is to add graduated non-convexity (GNC) (Nielsen, 1997; Zach & Bourmaud, 2018; Jin et al., 2024), which smooths the non-convex cost function by gradually reducing the scale parameter. Because it eliminates the competition from subpar solutions, it has shown to be the most promising strategy. In existing registration methods with GNC (Yang et al., 2020; Le & Zach, 2020; Gold & Rangarajan, 1996), parameter updates follow a basic and straightforward rule, multiplying by a given scaling factor constant during each iteration. The gradual optimization plan is carefully designed, which requires prior knowledge of the problem. An incorrect plan may lead to unnecessary long invalid runs in the registration instance. On the other hand, little attention has been paid to how the scaling factor is determined (Hazan et al., 2016; Le & Zach, 2020).

In this study, we introduce a robust point cloud registration method based on adaptive graduated non-convexity (AGNC). Different from previous GNC-based methods that rely on a predetermined update rule to adjust the shape of the cost function, we propose a new adaptive update rule to determine the scaling factor. The update rule aims to effectively adjust the shape of the cost function to minimize GNC iterations, thereby potentially improving the robustness of the method without sacrificing accuracy. To overcome the severe failure cases caused by high outlier rates, we propose a preventive measure. In the initial stage of AGNC, we achieve the co-optimization of non-convex cost functions at different levels through a multi-task knowledge sharing mechanism to jump out of the local minimum. This measure further reduces the failure rate of point cloud registration. Through performance evaluation on multiple datasets, we demonstrate the accuracy and robustness of AGNC to registration problems with outliers. Extremely high outlier percentages (such as 99% of correspondences being outliers) are acceptable to AGNC. To the best of our knowledge, this is the first study to explore point cloud registration considering adaptive graduated non-convexity.

The contributions of this work are as follows:

- We propose a novel approach to robust point cloud registration based on adaptive graduated non-convexity. The adaptive reduction of the graduated non-convexity scale occurs through monitoring the positive definiteness of the Hessian of the cost function.
- We achieve collaborative optimization of non-convex cost functions at different levels through a multi-task knowledge sharing mechanism to further improve the success rate of point cloud registration under challenging high outlier rates.
- Extensive experimental results on the different datasets demonstrate that our method can achieve superior registration precision and is robust to 99% outliers.

## 2 RELATED WORK

### 2.1 POINT CLOUD REGISTRATION

**Robust Methods.** RANSAC, as a well-known robust method, is embedded in the point cloud registration problem. It attempts to find reasonable samples and correctly identify them via iterations. Some methods perform preprocessing before RANSAC, considering the use of deterministic geometric methods (Bustos & Chin, 2017) or random game theory methods (Tam et al., 2012) to remove outliers. Potential outliers can also be selected for further processing, such as selecting potential inlier correspondences through geometric consistency checks (Barath & Matas, 2021). Some methods

perform transformation parameter search based on consensus maximization (Campbell et al., 2017) and Branch and Bound (BnB) techniques (Yang et al., 2021; Chen et al., 2022a). However, in the case of high outlier rates, all of the aforementioned techniques become intractable and accuracy is severely hampered.

**M-estimators.** The M-estimators method treats the point cloud registration problem as the minimization of a robust cost function. Cost functions include Geman-McClure (GM), Huber, Cauchy, Welsch, Tukey, etc. In the optimization, M-estimators give small weights (close to 0) to outliers and large weights (close to 1) to inliers. Therefore, the impact of outliers on the cost is largely discounted. (Zhou et al., 2016) proposed fast global registration (FGR), which uses the GM cost function and introduces Black-Rangarajan duality and GNC to solve the non-convex optimization problem. This duality provides a way to convert traditional line process methods and robust statistical methods into each other (Black & Rangarajan, 1996). In fact, when the proportion of outliers exceeds 80%, FGR tends to fail. (Enqvist et al., 2012) proposed sequential optimization of a range of surrogate functions instead of directly optimizing non-convex functions. GNC has achieved successful applications in computer vision (Black & Rangarajan, 1996; Nielsen, 1997; Zach & Bourmaud, 2018), and its wide applicability still needs to be explored.

**Deep Learning Methods.** The deep learning method first learns a high-dimensional feature space representation of the point cloud, then matches key points to generate hypothetical correspondences, and finally uses a differentiable registration module to obtain the best alignment (Wang et al., 2022; Yu et al., 2024; Liu et al., 2024; Wang et al., 2024). Many deep learning-based point cloud registration methods have been proposed, such as PointNetLK (Aoki et al., 2019), SpinNet (Ao et al., 2021) and FINet (Xu et al., 2022). The assumed correspondence can be obtained based on the features extracted from feature descriptors such as fully convolutional geometric features descriptor (FCGF) (Choy et al., 2019). For outliers in the hypothesized correspondences, some methods (Yu et al., 2021; Chen et al., 2022b; Qin et al., 2023; Mei et al., 2023) use spatial consistency metrics to eliminate outliers. Deep learning methods often have problems with generalization ability and the requirement for a large amount of training data.

## 2.2 GRADUATED NON-CONVEXITY

GNC is a commonly used method for optimizing non-convex cost functions and has been successfully applied in a variety of fields such as computer vision and machine learning (Black & Rangarajan, 1996; Nielsen, 1997). The fundamental idea of GNC is to continuously replace the original non-convex cost function with simpler functions, which leads to fewer local minima (Hazan et al., 2016; Yang et al., 2020). First, a simpler coarse-grained version of the objective is generated and minimized. Then, the version of the objective is gradually refined in stages, and the solution of the previous stage is used as the starting point for the optimization of the next stage. It eliminates the need for an initial guess and increases the probability of converging to the global minimum.

Let us explain GNC with an example. The GM function is a popular cost because of its robustness. The GM function and the surrogate function containing the scale parameter $\mu$ are as follows:

$$\rho(r) = \frac{\bar{c}^2 r^2}{2\left(\bar{c}^2 + r^2\right)} \implies \rho_\mu(r) = \frac{\mu \bar{c}^2 r^2}{2\left(\mu \bar{c}^2 + r^2\right)}, \tag{1}$$

where the parameter $c$ is assumed to be fixed, which controls the shape of $\rho(r)$. $\mu$ represents the scale of the noise, which distinguishes inliers and outliers. $r$ is the residual of the correspondence.

Fig. 1 shows a graphical representation of the cost function of $\rho_\mu(r)$ for different $\mu$ in GNC. The surrogate function $\rho_\mu(r)$ has the following characteristics: (i) $\rho_\mu(r)$ becomes convex for large $\mu$. (ii) $\rho_\mu(r)$ recovers $\rho(r)$ when $\mu = 1$. As the value of $\mu$ decreases, the cost function $\rho_\mu(r)$ starts to become non-convex and the number of local minima in the cost function landscape increases. GNC reduces $\mu$ to its final value $\mu_{final}$ by moving $r$ along the smooth red curve, which is the trajectory of the cost function minimum. At stage $k$, we estimate the minimum value $r_k$ at $\mu_k$.

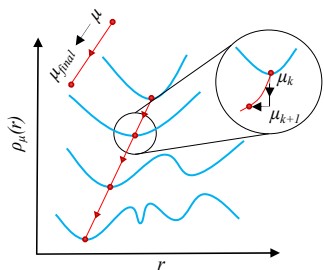

Figure 1: Cost function of $\rho_\mu(r)$ for different $\mu$ in GNC.

Then update the scale $\mu_k$ to $\mu_{k+1}$ and use $r_k$ as initialization to get the updated estimate $r_{k+1}$. The goal of the GNC technique is to guarantee that at each stage $(k+1)$, $r_k$ falls within the convergence region of the global minimum of the current cost function $\mu_{k+1}$. The ideal solution obtains the global minimum at the final $\mu_{final}$.

## 3 THE PROPOSED METHOD AGNC

### 3.1 PROBLEM FORMULATION

Finding a rotation matrix $\mathbf{R} \in SO(3)$ and a translation vector $\mathbf{t} \in \mathbb{R}^3$ that align a source point cloud $\mathbf{X}$ to a target point cloud $\mathbf{Y}$ is the aim of the point cloud registration. Given a set of correspondences $H = \{(x_i, y_i)\}_1^N$ with outliers, the problem of point cloud registration can be formulated as:

$$\min_{\mathbf{R} \in SO(3), \mathbf{t} \in \mathbb{R}^3} \sum_{i=1}^{N} \rho_\mu \left( \|\mathbf{R}\mathbf{x}_i + \mathbf{t} - \mathbf{y}_i\| \right), \tag{2}$$

where the notation $\| \cdot \|$ represents the $l_2$-norm, and $\rho_\mu$ is a robust cost function. When $\mu \to \infty$, the registration problem can be estimated by the least squares method, that is,

$$\min_{\mathbf{R} \in SO(3), \mathbf{t} \in \mathbb{R}^3} \frac{1}{2} \sum_{i=1}^{N} \|\mathbf{R}\mathbf{x}_i + \mathbf{t} - \mathbf{y}_i\|^2. \tag{3}$$

It can find the global minimum by Umeyama method (Umeyama, 1991). For other values of $\mu$, it will lead to a weighted least squares problem:

$$\min_{\mathbf{R} \in SO(3), \mathbf{t} \in \mathbb{R}^3} \frac{1}{2} \sum_{i=1}^{N} w_i \|\mathbf{R}\mathbf{x}_i + \mathbf{t} - \mathbf{y}_i\|^2. \tag{4}$$

It can also be solved by the weighted Umeyama method (Umeyama, 1991).

### 3.2 ADAPTIVE GRADUATED NON-CONVEXITY

Although GNC has been successful in early computer vision applications, most of them use a simple fixed update rule (Nielsen, 1997; Ochs et al., 2013; Hazan et al., 2016; Yang et al., 2020). The scale $\mu$ is decreased by a predetermined step size at each iteration, that is, $\mu_{k+1} = \frac{\mu_k}{\zeta}$, where $\zeta > 1$. The performance of GNC depends critically on the update method used for the scale parameter $\mu$. Imagine that if $\zeta$ is close to 1, the movement in the cost function landscape becomes slow. This conservative strategy ensures that each step in the optimization process moves firmly along the red curve and finally reaches the global minimum at $\mu_{final}$. However, this method requires a large number of update stages to gradually reduce $\mu$, which undoubtedly increases the computational cost of the entire optimization process. In contrast, if we choose a larger value of $\mu$, the movement in the cost function landscape becomes very fast. But this fast-moving strategy also brings the risk that the algorithm may not fully explore all areas in the cost landscape and get stuck in a local minimum.

In this paper, we propose a robust point cloud registration method with adaptive graduated non-convexity. At each stage, we seek to use the largest $\mu$ possible while ensuring that each step update of the algorithm lies within the expected convergence range of the global minimum, significantly improving the accuracy and reliability of point cloud registration. To accomplish this, we look at the Hessian of the cost function Eq. 2.

$$[\mathbf{H}_i]_{(r,s)} = \left. \frac{\partial^2 \rho_\mu \left( \|\mathbf{r}_i(z)\| \right)}{\partial z_r \partial z_s} \right|_{z_k}, \tag{5}$$

where $z$ is the estimated parameter $\mathbf{R}$ and $\mathbf{t}$, $\mathbf{r}_i(z)$ is the $i$-th corresponding residual value. The partial derivatives of $z_r$ and $z_s$ are with respect to the two components of rotation and translation.

Since $z_k$ is the minimum of the cost function evaluated for $\mu_k$ in Eq. 2, $\mathbf{H}$ is locally convex, i.e. positive definite. In the $k+1$ stage, when the scale is updated to $\mu_{k+1}$, if the corresponding Hessian $\mathbf{H}$ in Eq. 2 obtained at $\mu_{k+1}$ is ensured to remain positive definite, then the new estimate $z_{k+1}$ is guaranteed to be in the same convergence domain as the previous iteration (Andrew & Gao, 2007;

Koh et al., 2007; Ochs et al., 2013). The solution $z_{final}$ obtained in this way is likely to be the global minimum at $\mu_{final}$.

$\mathbf{H}$ is positively definite with all positive eigenvalues, and its positive definiteness can be ensured by keeping track of the sign of the smallest eigenvalue $\lambda_{min}$ of $\mathbf{H}$. The condition for preserving local convexity translates to finding the minimum $\mu_{k+1}$ while keeping $\lambda_{min}(\mathbf{H}) > 0$ at each iteration. We exclusively determine $\mu_{k+1}$ based on the criterion of $\lambda_{min}(\mathbf{H}) > 0$, and we never employ $\mathbf{H}$ in the estimating process, despite the fact that $\lambda_{min}(\mathbf{H})$ close to zero renders $\mathbf{H}$ exceedingly ill-conditioned. In addition, we have the option to stop the search when $\lambda_{min}(\mathbf{H})$ gets close to a threshold, ensuring that $\mathbf{H}$ is never ill-conditioned. We emphasize again that $\mathbf{H}$ is only used to find $\mu_{k+1}$ and not in the optimization step. This adaptive update step of $\mu$ not only improves the optimization efficiency but also has no detrimental effects on solution accuracy.

At the point $(\mathbf{R}, \mathbf{t})$, the Hessian $\mathbf{H}$ of the Eq. 2 is

$$\mathbf{H} = \sum_{i=1}^{N} \mathbf{H}_i = \sum_{i=1}^{N} \left( -l_i \frac{\mathbf{g}_{LSQ,i}\mathbf{g}_{LSQ,i}^{\top}}{\|\mathbf{r}_i\|^2} + m_i \mathbf{H}_{LSQ,i} \right), \tag{6}$$

$$\mathbf{g}_{LSQ,i} = \left[ \begin{array}{c} -\left[\mathbf{x}_i\right]_{\times} \mathbf{R}^{\top} \mathbf{r}_i \\ -\mathbf{r}_i \end{array} \right], \tag{7}$$

$$\mathbf{H}_{LSQ,i} = \left[ \begin{array}{cc} \left(\mathbf{p}_i^{\top}\mathbf{R}\mathbf{x}_i\right)\mathbf{I} - \frac{\mathbf{x}_i\mathbf{p}_i^{\top}\mathbf{R}}{2} - \frac{\mathbf{R}^{\top}\mathbf{p_i}\mathbf{x}_i^{\top}}{2} & \left[\mathbf{x}_i\right]_{\times}\mathbf{R}^{\top} \\ -\mathbf{R}\left[\mathbf{x}_i\right]_{\times} & \mathbf{I} \end{array} \right], \tag{8}$$

where the residual of the $i$-th correspondence $r_i = \mathbf{R}\mathbf{x}_i + \mathbf{t} - \mathbf{y}_i$ and $\mathbf{p}_i = \mathbf{y}_i - \mathbf{t}$. $\mathbf{I}$ is the $3 \times 3$ identity matrix. $\mathbf{g}_{LSQ,i}$ is the gradient and $\mathbf{H}_{LSQ,i}$ is the Hessian of the least squares cost at the $i$-th residual. $l_i$ and $m_i$ are factors for weight adjustment based on the residuals, which are used to modify the gradient and Hessian. $x_i$ is the coordinate of the $i$-th point in the source point cloud $\mathbf{X}$. $[\ ]_{\times}$ is a skew-symmetric matrix operation, which converts a coordinate vector into a skew-symmetric matrix form for cross-multiplication with the rotation vector. $\top$ is the transpose of a matrix.

This principle scheme is universal, and we still take the GM cost function as an example. We have

$$l_i = \frac{4\|\mathbf{r}_i\|^2}{\mu^2\left(1 + \frac{\|\mathbf{r}_i\|^2}{\mu^2}\right)^3}, \quad m_i = \frac{1}{\left(1 + \frac{\|\mathbf{r}_i\|^2}{\mu^2}\right)^2}. \tag{9}$$

Then, the Hessian $\mathbf{H}$ is

$$\mathbf{H} = \sum_{i=1}^{N} \frac{-4\mathbf{g}_{LSQ,i}\mathbf{g}_{LSQ,i}^{\top}}{\mu^2\left(1 + \frac{\|\mathbf{r}_i\|^2}{\mu^2}\right)^3} + \frac{1}{\left(1 + \frac{\|\mathbf{r}_i\|^2}{\mu^2}\right)^2}\mathbf{H}_{LSQ,i}. \tag{10}$$

In general, we usually cannot obtain a closed-form expression for $\lambda_{min}(\mathbf{H})$. Therefore, we use a divide-and-conquer approach to estimate $\mu_{k+1}$ based on the condition that $\lambda_{min}(\mathbf{H}) > 0$. We do a binary search with a search interval defined below $\mu_k$. The binary search strategy is based on an implicit assumption that $\lambda_{min}$ will decrease monotonically as $\mu$ is gradually reduced. We can further decrease the search interval to make sure this assumption is reliable. Since $\mathbf{H}$ is a small $6 \times 6$ matrix, so the cost of evaluating $\lambda_{min}(\mathbf{H})$ is low. Although the cost function of Eq. 2 is nonlinear, it is smooth and differentiable.

In Fig. 2, we show the effectiveness of adaptive GNC in dealing with a simple 2D linear fitting problem with outliers. Table 1 lists the comparison of the convergence stages under different annealing strategies and the quality of the final solution. It takes 16 stages for GNC to converge to the global minimum when $\zeta$ is set to a small value ($\zeta = 4$). In contrast, if a larger $\zeta$ is used ($\zeta = 20$), the GNC optimization terminates after only 6 stages but often falls into suboptimal local minima. In sharp contrast, the proposed adaptive

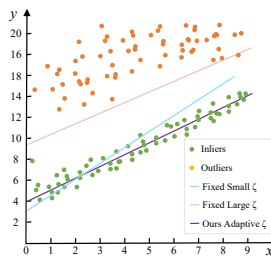

Figure 2: Example of adaptive GNC for a line fitting problem with outliers present.

GNC successfully converges to the global minimum after 8 stages, resulting in the correct fitting solution. Ours achieves the highest accuracy with faster convergence and less overhead. We also provide a comparison of AGNC with two scale adaptive schemes GradOpt (Hazan et al., 2016) and ASKER (Le & Zach, 2020) in Table 2 of the supplementary material.

Table 1: Comparison of different updating methods of $\zeta$, time unit is in ms.

| GNC strategy | Stages | Runtime | Hessiantime | Accuracy |
|---|---|---|---|---|
| Fixed small $\zeta$ | 16 | 4.98 | - | Medium |
| Fixed large $\zeta$ | 6 | 2.02 | - | Low |
| Ours adaptive $\zeta$ | 8 | 3.61 | 1.08 | High |

### 3.3 MULTI-TASK KNOWLEDGE SHARING

To overcome the severe failure cases caused by high outliers, we propose a preventive measure. Inspired by human learning, humans often use their experience of solving one problem to help solve other problems (Chen et al., 2018; Xu et al., 2020). Improve the optimization performance of multiple related tasks by sharing knowledge between tasks (Gupta et al., 2015; Liao et al., 2023; Yang et al., 2023). We regard the cost functions at different stages of the optimization process as different tasks, whose function landscapes or optimal solutions have certain similarities. A promising candidate solution that helps on one task may also help on another task. Therefore, in the initial stage of AGNC, we implement the collaborative optimization of non-convex cost functions at different levels through a multi-task sharing mechanism to jump out of the local minimum. This measure further improves the success rate of point cloud registration under challenging high outliers. The multiple AGNC optimization problem can be expressed as:

$$\arg\min \left\{ f_{\mu_k}(z), f_{\mu_{k-1}}(z), \ldots, f_{\mu_{k-j}}(z) \right\}. \tag{11}$$

### 3.4 FRAMEWORK OF AGNC FOR POINT CLOUD REGISTRATION

The pseudo-code of the adaptive GNC for the point cloud registration problem is shown in Algorithm 1. The input point cloud correspondence includes outliers. Calculate the current residual $r_i$ based on $N$ sets of correspondences (line 2). According to the residual $r_i$, calculate the weight $w_i$ (line 3). Using the weighted Umeyama method, the rotation matrix $\mathbf{R}$ and the translation vector $\mathbf{t}$ are solved according to the weight $w_i$ (line 4). A multi-task knowledge sharing strategy is implemented to achieve joint optimization of non-convex cost functions at different levels to prevent falling into local minima (line 5). Calculate the Hessian matrix $\mathbf{H}$ and perform a binary search on the minimum eigenvalue $\lambda_{\min}(\mathbf{H})$ of $\mathbf{H}$ to obtain $\mu_{k+1}$ (line 6-7). When $\mu_k$ reaches the threshold $\mu_{final}$, the iterative process ends. Compared to traditional fixed-step optimization plans, the scale of graduated non-convexity is adaptively reduced by monitoring the positive definiteness of the Hessian of the cost function.

---

**Algorithm 1** Point cloud registration based on AGNC

---

**Input:** $H = \{(x_i, y_i)\}_1^N$ with outliers in the two point clouds, $\mu_{final}$, $k = 0$, $\mu = \mu_0$
**Output:** Rotation matrix $\mathbf{R}$, translation vector $\mathbf{t}$
1: **while** $\mu_k \geq \mu_{final}$ **do**
2:     $r_i = \mathbf{R}\mathbf{x}_i + \mathbf{t} - \mathbf{y}_i$
3:     $w_i = \frac{1}{\left(1 + \frac{\|\mathbf{r}_i\|^2}{\mu_k^2}\right)^2}$
     /* find $\mathbf{R}$ and $\mathbf{t}$ by weighted Umeyama method */
4:     $\mathbf{R}, \mathbf{t} = \text{WeightedUmeyama}\left(\{(x_i, y_i, w_i)\}_1^N\right)$
5:     Perform multi-tasking knowledge sharing
6:     Calculate the $\mathbf{H}(\mu)$ using Eq. 10
7:     Run binary search on $\lambda_{\min}(\mathbf{H})$ to obtain $\mu_{k+1}$
8:     $k = k + 1$
9: **end while**

---

## 4 EXPERIMENTS

### 4.1 DATASETS AND COMPARING METHODS

The experiments consider four point cloud registration datasets. The Stanford repository (Curless & Levoy, 1996) contains four object models, i.e., Bunny, Dragon, Armadillo, and Buddha, which are used for simulation experiments. 3DMatch (Zeng et al., 2017) and 3DLoMatch (Huang et al., 2021) are two indoor scene datasets. 3DLoMatch is a subset of 3DMatch, where the overlap rate of point cloud pairs is between 10% and 30%. Registration under high outliers is very challenging. KITTI (Geiger et al., 2012) is a large-scale outdoor scene dataset. More details of the datasets are reported in Table 1 of the supplementary material.

We compare our method AGNC with eight representative point cloud registration methods, the classic RANSAC (Fischler & Bolles, 1981) and its variant GC-RANSAC (Barath & Matas, 2021), the fast global registration method FGR (Zhou et al., 2016), TEASER++ (Yang et al., 2021) a GNC-based method with a fixed update rule, SC$^2$-PCR (Chen et al., 2022b), TR-DE (Chen et al., 2022a) and HERE (Huang et al., 2024) through transformation parameter decomposition search, and MAC(Zhang et al., 2023) using maximal cliques to prune outliers. The source code can be found in their respective papers. For AGNC, we fix $\mu_{final} = 0.1$ unless otherwise stated. All statistics are calculated using 100 Monte Carlo runs.

### 4.2 EVALUATION METRICS

Following (Yang et al., 2021), we employ rotation error $RE$ and translation error $TE$ to evaluate the registration performance, which are shown below:

$$RE = \arccos\left(\frac{\mathrm{Tr}(\mathbf{R}_{gt}^{\mathrm{T}}\mathbf{R}^*) - 1}{2}\right), \tag{12}$$

$$TE = \|\mathbf{t}_{gt} - \mathbf{t}^*\|, \tag{13}$$

where $\mathrm{Tr}(\cdot)$ is the trace of a matrix. $\mathbf{R}^*$ and $\mathbf{t}^*$ are estimated values. $\mathbf{R}_{gt}$ and $\mathbf{t}_{gt}$ are ground truth values. The lower the values of these two indicators, the better the method.

We also report the registration recall $RR$ for real-world datasets, which refers to the proportion of successful registrations with $RE$ error and $TE$ error falling within predetermined bounds.

$$RR = \frac{\text{\# successful registration instance}}{\text{\# all registration instance}}. \tag{14}$$

### 4.3 COMPARISON ON SIMULATED DATASETS

We first conduct experiments on simulated data from the Stanford repository to validate our proposed method. We create an outlier simulated dataset as suggested in TEASER++ (Yang et al., 2021). Specifically, the input outlier contaminated correspondences $H = \{(x_i, y_i)\}_1^N$ are generated as follows: First, the original point cloud is downsampled to $N = 2000$ points and resized to fit into $[0, 1]^3$ to create the source point cloud $\mathbf{X}$. Then, the $\mathbf{X}$ is transformed to another local coordinate system by transforming $\mathbf{R}x_i + \mathbf{t} - y_i$ to obtain the target point cloud $\mathbf{Y}$, where the rotation matrix $\mathbf{R}$ is a randomly generated 3×3 Rodrigues matrix ($\mathbf{R} \in SO(3)$) and the translation $\mathbf{t}$ is a randomly generated 3×1 vector ($0 \leq \|\mathbf{t}\| \leq 1$). To simulate the noise present in real data, we add random bounded noise $\boldsymbol{\epsilon_i} \sim \mathcal{N}\left(\mathbf{0}, \eta^2\mathbf{I}\right)$ to $\mathbf{Y}$ $\left(\|\boldsymbol{\epsilon}_i\|^2 \leq \beta_i\right)$ with $\beta_i = 5.54\eta, \eta = 0.01$ as chosen in (Yang et al., 2021). To generate outlier correspondences, a certain percentage of points $Y$ are randomly selected and replaced by vectors uniformly sampled within a sphere with a radius of 8 units. The level of outliers is measured by the number of wrong correspondences and the ratio of all correspondences. The outlier level is set to 0%, 20%, 40%, 60%, 80%, 90%, and 99%. Fig. 3 shows the rotation error and translation error of compared methods at different outlier levels.

From the results, we can see that when the outlier level is low, all methods perform similarly. As the outlier level increases, the errors of some methods (RANSAC, GC-RANSAC, and FGR) increase significantly. RANSAC, GC-RANSAC, and FGR perform poorly at extreme outlier rates.

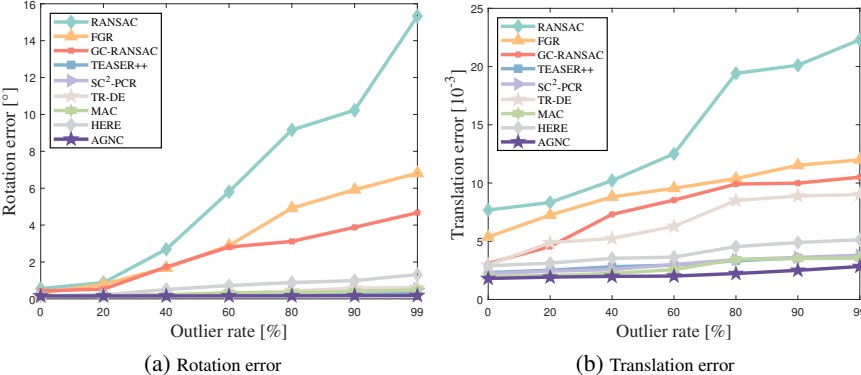

(a) Rotation error

(b) Translation error

Figure 3: Rotation and translation error with increasing outlier rates on the Stanford repository.

TEASER++, SC$^2$-PCR, TR-DE, MAC, and AGNC are robust to outliers up to 99%. Although they are all robust to 99% outliers, AGNC produces a lower estimation error. The experimental results show that our method can effectively handle point cloud registration problems with different degrees of outliers.

Table 2 reports the quantitative results of all methods at 50% outlier rate. Our method achieves the best performance on all models, i.e., the best RE, TE, and RR. The visual registration results of the AGNC method at 50% outlier rate are shown in Fig. 4. The first, second, and third rows show the input, the ground truth, and the AGNC registration results, respectively. For more visualizations of comparisons, please see the supplementary material. From a visual perspective, our method shows excellent registration performance on all models. It is close to the true value and no obvious registration deviation is observed. This further verifies the accuracy and reliability of our method on the registration problem with outliers. Please see Fig. 1 in the supplementary material for more visual results.

Table 2: Registration results with 50% outliers rate on Stanford repository.

| Method | Bunny | Dragon | Armadillo | Buddha | Bunny | Dragon | Armadillo | Buddha | Avg. |
|---|---|---|---|---|---|---|---|---|---|
| | Rotation Errors(deg)↓ | | | | Translation Errors($\times 10^{-3}$)↓ | | | | RR(%)↑ |
| RANSAC | 11.76 | 10.83 | 9.37 | 4.67 | 22.1 | 18.64 | 17.55 | 19.37 | 59.42 |
| FGR | 5.37 | 4.29 | 5.11 | 4.91 | 11.37 | 8.3 | 9.83 | 12.01 | 68.32 |
| GC-RANSAC | 3.16 | 2.38 | 3.55 | 3.37 | 8.61 | 8.44 | 9.08 | 13.50 | 75.19 |
| TEASER++ | 0.59 | 0.65 | 0.60 | 0.35 | 2.38 | 2.55 | 2.22 | 2.53 | 96.75 |
| SC$^2$-PCR | 0.33 | 0.38 | 0.47 | 0.35 | 4.61 | 3.05 | 3.95 | 2.08 | 95.10 |
| TR-DE | 0.73 | 0.60 | 0.55 | 0.42 | 9.61 | 7.68 | 8.92 | 7.78 | 84.79 |
| MAC | 0.53 | 0.46 | 0.50 | 0.38 | 3.11 | 3.08 | 3.93 | 3.64 | 95.86 |
| HERE | 0.85 | 0.83 | 0.87 | 0.99 | 5.91 | 6.91 | 5.34 | 5.70 | 87.61 |
| AGNC (Ours) | **0.19** | **0.15** | **0.14** | **0.18** | **2.05** | **2.42** | **2.11** | **2.32** | **98.94** |

### 4.4 COMPARISON ON REAL-WORLD DATASETS

**Evaluation on Indoor Scenes.** First, we consider the 3DMatch dataset, which contains 62 real indoor scenes. It is divided into 54 scenes for training and 8 scenes for testing. Features are obtained from FCGF and FPFH descriptor (Chen et al., 2022b), then matched using nearest-neighbor matching. In the correspondences, the outlier percentage varies from 0% to 99%. Therefore, some registration instances are bound to fail. We use the same successful registration criteria defined in (Zhang et al., 2023; Chen et al., 2022b; Huang et al., 2024), namely $RE \leq 15°$ and $TE \leq 30cm$ relative to the ground truth. As can be seen from Table 3, AGNC has a lower rotation error and translation error compared to other methods. In addition, the registration recall of AGNC is still 0.15 higher than the highest method MAC. More results with different ouliers can be found in Fig. 2 of the supplementary material. Next, we conducted experiments on the 3DLoMatch dataset. 3DLo-Match has a lower overlap rate than 3DMatch point clouds. The experimental setting follows (Chen et al., 2022b;a), using the Predator and FCGF descriptor to generate the initial correspondence set. From the results in Table 4, it can be observed that our method achieves the highest successful alignment percentage together with SC$^2$-PCR. However, our method achieves better performance in

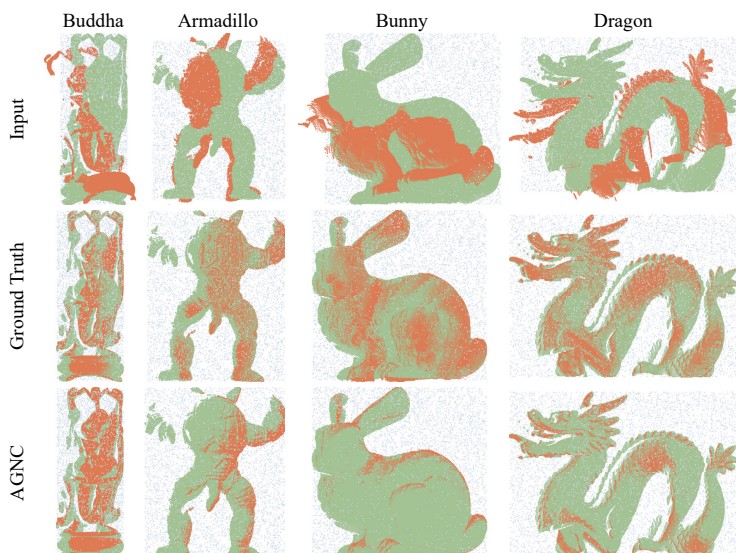

Figure 4: Visualization results with 50% outliers rate on the Stanford repository.

Table 3: Registration results on the 3DMatch dataset.

| Method | FCGF Descriptor | | | FPFH Descriptor | | |
|---|---|---|---|---|---|---|
| | RR(%)↑ | RE(°)↓ | TE(cm)↓ | RR(%)↑ | RE(°)↓ | TE(cm)↓ |
| RANSAC | 89.22 | 2.46 | 7.60 | 64.20 | 4.05 | 11.35 |
| FGR | 73.75 | 2.73 | 8.14 | 40.91 | 4.96 | 10.25 |
| GC-RANSAC | 89.65 | 2.36 | 7.23 | 67.65 | 2.33 | 6.87 |
| TEASER++ | 85.77 | 2.91 | 9.40 | 75.48 | 2.48 | 7.31 |
| SC2-PCR | 92.73 | 2.20 | 6.88 | 83.98 | 2.18 | 6.56 |
| TR-DE | 86.99 | 2.62 | 8.03 | 77.18 | 2.89 | 8.83 |
| MAC | 92.79 | 2.18 | 6.89 | 84.10 | 1.96 | 6.18 |
| HERE | 91.56 | 2.17 | 6.93 | 83.08 | 2.94 | 7.02 |
| AGNC | **92.94** | **2.03** | **6.56** | **84.12** | **1.94** | **6.18** |

terms of rotation error and translation error. This shows that the alignment of the AGNC method is very accurate and can align low-overlapping data. Refer to Fig. 3 in the supplementary material for visual results on the indoor scenes.

**Evaluation on Outdoor Scenes.** We complete outdoor scenes registration tests on the KITTI dataset. Following (Chen et al., 2022b), we use the 8th to 10th scenes to evaluate all methods. For the assumed correspondences, we use the FPFH descriptor (Rusu et al., 2009) and the FCGF descriptor (Choy et al., 2019) to generate the initial correspondence set, respectively. We set the thresholds to $RE \leq 5°$ and $TE \leq 60cm$ as the criteria for evaluating $RR$. The experimental re-

Table 4: Registration results on the 3DLoMatch dataset.

| Method | Predator Descriptor | | | FCGF Descriptor | | |
|---|---|---|---|---|---|---|
| | RR(%)↑ | RE(°)↓ | TE(cm)↓ | RR(%)↑ | RE(°)↓ | TE(cm)↓ |
| RANSAC | 66.03 | 3.76 | 11.82 | 46.38 | 5.00 | 13.11 |
| FGR | 38.90 | 3.90 | 11.63 | 19.99 | 5.28 | 12.98 |
| GC-RANSAC | 64.18 | 3.39 | 11.21 | 48.62 | 4.21 | 10.72 |
| TEASER++ | 63.17 | 4.17 | 10.58 | 46.76 | 4.12 | 12.89 |
| SC2-PCR | 68.73 | 3.22 | 10.75 | 57.83 | 3.77 | 10.92 |
| TR-DE | 66.03 | 4.32 | 11.04 | 49.50 | 4.46 | 12.07 |
| MAC | 69.17 | 3.42 | 10.47 | 59.85 | 3.50 | 9.75 |
| HERE | 68.89 | 3.31 | 10.42 | 57.08 | 3.48 | 10.81 |
| AGNC | **69.17** | **3.19** | **9.98** | **59.89** | **3.45** | **9.73** |

sults are listed in Tables 5. The $RE$ and $TE$ of AGNC are lower than those of the state-of-the-art heuristic-guided parameter search method HERE. It can be concluded that AGNC outperforms all the compared methods regardless of the descriptor used. AGNC achieves the best $RR$, $RE$, and $TE$ indicators, indicating its strong registration ability for outdoor scene point clouds. AGNC's strong generalization capacity across many application scenarios is confirmed by registration studies conducted on object, indoor scene, and outdoor scene datasets. Refer to Fig. 4 in the supplementary material for visual results on the outdoor scenes.

Table 5: Registration results on the KITTI dataset.

| Method | FPFH Descriptor | | | FCGF Descriptor | | |
|---|---|---|---|---|---|---|
| | RR(%)↑ | RE(°)↓ | TE(cm)↓ | RR(%)↑ | RE(°)↓ | TE(cm)↓ |
| RANSAC | 95.67 | 1.06 | 23.19 | 98.01 | 0.39 | 21.73 |
| FGR | 9.73 | 0.58 | 27.84 | 97.47 | 0.34 | 19.86 |
| GC-RANSAC | 79.46 | 0.39 | 8.02 | 97.47 | 0.32 | 20.50 |
| TEASER++ | 97.84 | 0.43 | 8.39 | 98.02 | 0.34 | 20.74 |
| SC$^2$-PCR | 99.64 | 0.39 | 8.29 | 97.66 | **0.31** | 20.21 |
| TR-DE | 98.91 | 0.92 | 15.63 | 97.11 | 0.83 | 24.33 |
| MAC | 99.10 | 0.51 | 10.17 | 97.66 | 0.45 | 23.40 |
| HERE | 99.10 | 0.42 | 7.90 | 98.02 | 0.32 | 20.73 |
| AGNC | **99.71** | **0.32** | **7.25** | **98.52** | **0.31** | **19.51** |

To verify the impact of the two key strategies, ablation studies are conducted following the experimental design of simulated datasets. The fixed update scheme uses $\zeta = 1.5$ and tests the effect of no multi-task knowledge transfer. The results are shown in Table 6. For all four models, our overall design produces lower rotation error and translation error. The reason is that tracking local minima sometimes leads to solutions far away from the ground truth due to the challenge of high outliers. At this time, multi-task sharing mechanisms are needed to learn other levels of non-convex cost function landscapes to jump out of local minima. Adaptive graduated non-convexity effectively adjusts the shape of the cost function according to the optimization process to enhance the registration's accuracy and robustness.

Table 6: Ablation study of two key strategies. S/U are stages and runtime (ms) respectively.

| Dataset | Fixed w/ sharing | | | Adaptive w/o sharing | | | Fixed w/o sharing | | | AGNC | | |
|---|---|---|---|---|---|---|---|---|---|---|---|---|
| | RE | TE | S/U | RE | TE | S/U | RE | TE | S/U | RE | TE | S/U |
| Bunny | 0.88 | 2.53 | 12/13.68 | 2.34 | 5.85 | 5/5.23 | 3.33 | 6.01 | 12/12.34 | 0.19 | 2.05 | 6/7.32 |
| Dragon | 0.56 | 2.91 | 11/12.71 | 5.18 | 12.99 | 5/5.23 | 5.39 | 13.02 | 11/11.29 | 0.15 | 2.42 | 5/6.34 |
| Armadillo | 0.42 | 2.67 | 12/13.68 | 1.58 | 3.64 | 5/5.23 | 3.28 | 5.11 | 12/12.34 | 0.14 | 2.11 | 5/6.34 |
| Buddha | 0.49 | 3.01 | 10/11.64 | 2.93 | 4.08 | 6/6.26 | 3.64 | 5.83 | 10/10.21 | 0.18 | 2.32 | 6/7.32 |

## 5 CONCLUSION

We have proposed a novel robust point cloud registration approach based on adaptive graduated non-convexity. Without requiring a set optimization plan, the scale of graduated non-convexity is adaptively lowered by keeping an eye on the positive definiteness of the Hessian of the cost function. Experimental results have shown that this method outperforms compared methods in terms of robustness and accuracy, can obtain promising registration results even in 99% outlier rates.

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
