# ADAPTIVE GRADUATED NON-CONVEXITY FOR POINT CLOUD REGISTRATION - SUPPLEMENTARY MATERIAL

## 1 EXPERIMENTAL DATASETS

The detailed information of experimental datasets used in this paper is shown in Table 1. These datasets are selected according to different criteria, such as scene complexity, acquisition technology, and data quality. (i) Diverse application scenarios. Three typical application scenarios of point cloud registration are covered, namely object models, indoor scenes, and outdoor scenes. (ii) Different data modalities. Since sensors may vary with different applications, different sensor technologies will produce point clouds of different qualities. For example, there is real noise in Kinect captured data. Therefore, it is necessary to consider the performance differences of registration methods under different data modalities. (iii) Rich overlap ratios. In point cloud registration, data scanned from different angles usually have limited overlap. It naturally leads to many correspondences falling outside the overlapping area. (iv) Different initial misalignment. The rich initial rotation errors and translation errors put forward higher requirements on the convergence of the algorithms.

Table 1: Detailed information of experimental datasets.

| Dataset | Modality | Challenge | Scenario | Rotation | Translation |
|---|---|---|---|---|---|
| Stanford | Kinect | | Object | 78.62 | 149.04 |
| 3DMatch | LiDAR | Initial Misalignment, | Indoor Scenes | 80.32 | 185.63 |
| 3DLoMatch | LiDAR | Noise, Outlier, Limited Overlap | Indoor Scenes | 83.49 | 197.08 |
| KITTI | LiDAR | | Outdoor Scenes | 79.51 | 298.74 |

## 2 VISUALIZATION OF EXPERIMENTAL RESULTS

The visualization results of all methods in the Stanford repository are shown in Figure 1. Visually, no obvious registration deviation is observed of our AGNC method on all models. The registered point clouds are well matched and basically consistent with the ground-truth registration results.

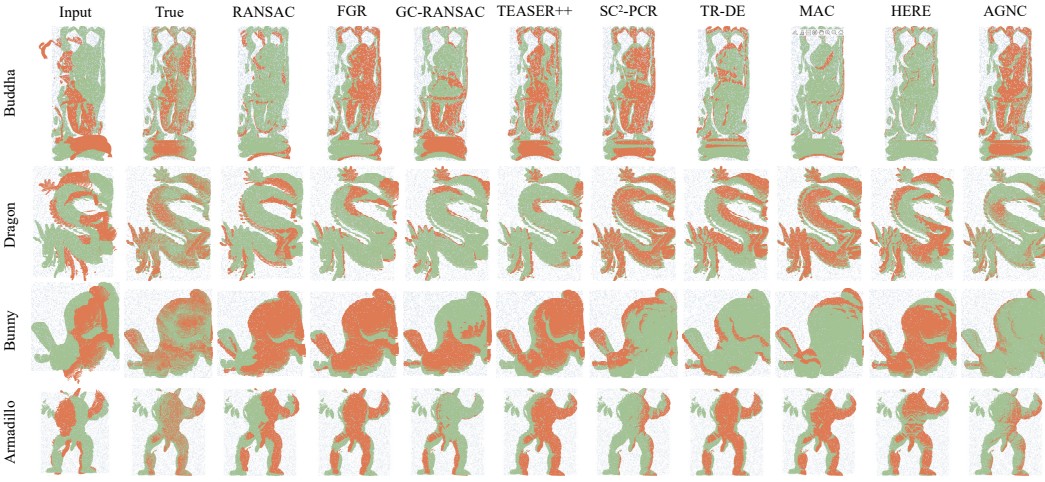

Figure 1: Visualization results of all methods with 50% outliers rate on the Stanford repository.

Figure 2 shows the rotation error and translation error of our AGNC method under different outlier levels. From the results, it can be seen that with the increase of outlier level, the registration error gradually becomes larger. Even in the case of higher outliers, the error is still within an acceptable range.

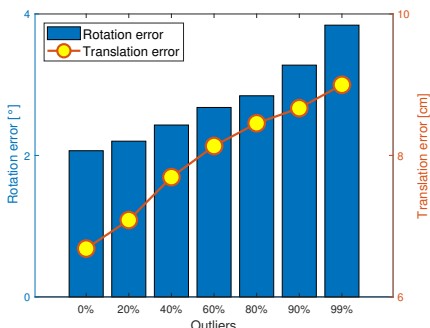

Figure 2: Rotation and translation error of AGNC with various outlier rates on the 3DMatch dataset.

The visual results of point cloud registration by AGNC for indoor and outdoor scenes are shown in Figures 3 and 4. The first and second rows show the state before registration and the ground truth, respectively. The last row shows the registration results of the AGNC method. No significant changes are observed in all the registration examples by our method. The registration results are basically consistent with the ground-truth results.

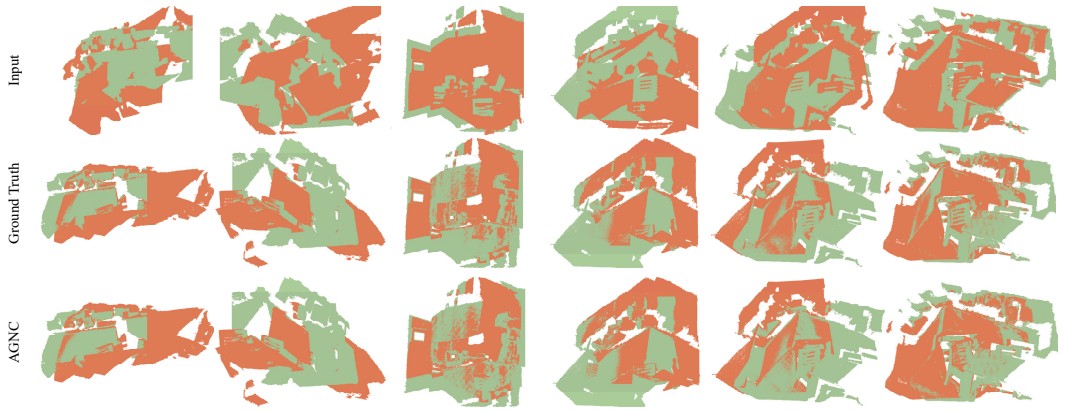

Figure 3: Visualization results of AGNC on the indoor scenes.

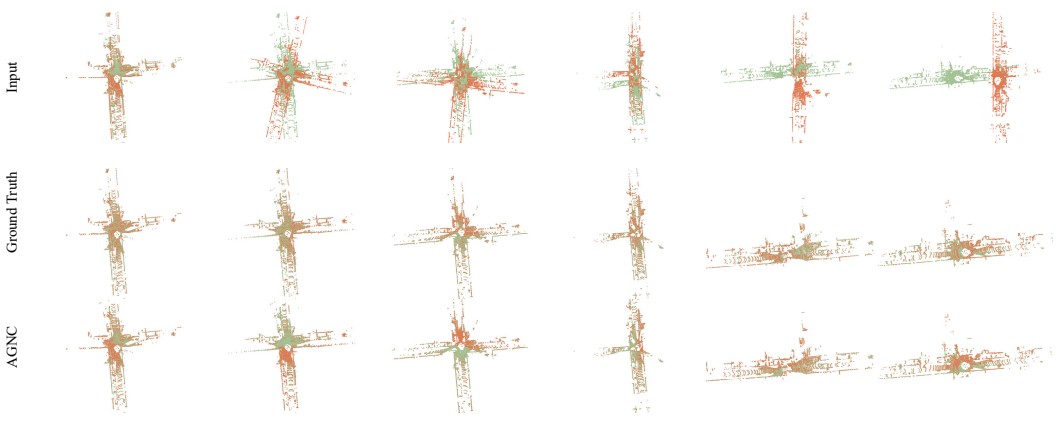

Figure 4: Visualization results of AGNC on the outdoor scenes.

## 3 COMPARE WITH OTHER SCALING SCHEMES

Our method is compared with two different scaling schemes, GradOpt (Hazan et al., 2016) and ASKER(Le & Zach, 2020). Table 2 reports the results of convergence steps, runtime, and relative accuracy on the Stanford dataset. The results show that our AGNC achieves the best results in both speed and accuracy. Compared with the second-ranked ASKER, the convergence steps are increased by 11 stages, the runtime is improved by 3.92 and with higher accuracy.

Table 2: Compare with different scaling schemes, time unit is in ms.

| Method | Stages | Runtime | Accuracy |
|--------|--------|---------|----------|
| GradOpt | 40 | 18.46 | Low |
| ASKER | 17 | 10.65 | Medium |
| AGNC | 6 | 6.73 | High |

## 4 DISCUSSION OF SPECIFIC APPLICATIONS

Point cloud registration technology plays a vital role in the application fields of virtual reality and self-driving cars. However, the requirements and tolerances for registration accuracy vary in different application scenarios.

In virtual reality application scenarios, such as room-scale virtual reality, the goal is usually to build an overall model of the space rather than precise positioning. In this case, large translation errors after registration are usually acceptable and have little impact on the user experience. They will not be noticed in large space scenes, so they do not affect the immersion. In the environmental modeling of autonomous vehicles, errors within a tolerable range will not have a significant impact on the vehicle's path planning and decision-making. In urban environments, autonomous driving systems rely on real-time data updates, multi-sensor registration and fusion technology to correct positioning. They work together to provide continuous and high-precision position information. Therefore, registration errors within a tolerable range will not affect the safe driving of the vehicle. Of course, significant errors may result in critical positioning deviations, causing the vehicle to deviate from the road, fail to follow the lane, or even collide with traffic lights.