# OpenReview forum: "Adaptive Graduated Non-Convexity for Point Cloud Registration"
_ICLR.cc/2025/Conference — ICLR 2025 Conference Desk Rejected Submission_

### Official Review · Reviewer_k1rY · 2024-11-01

**Soundness:** 3
**Presentation:** 2
**Contribution:** 3
**Rating:** 6
**Confidence:** 3

**Summary:**

This paper focuses on the task of registration, specifically on estimating the rotation and translation given correspondences that may exist outliers. To this end, it presents a method to register point clouds rubostly by leveraging Adaptive Graduated Non-Convexity (AGNC). First, the scale in graduated non-convexity is reduced by monitoring the positive definiteness of the Hessian of the cost function. Then knowledge sharing mechanism across different tasks is used to achieve collaborative optimiazation. Experiments are conducted on both synthetic and real datasets, and the results demonstrate the significance of the proposed method for robust point cloud registration.

**Strengths:**

1. The idea is novel, and from the experimental results, it helps boost the registration performance without additional traning procedure. It can be widely adopted as an alternative to RANSAC;

2.The overall writing is OK

3. Experiments are conducted on both synthetic and real data, and the results are convincing

**Weaknesses:**

1. In the experimental results, it seems there is no time analysis. It is better to compare the proposed method to existing ones in terms of time cost;

2. The experiments on Indoor Scenes may be insufficient, as this is the main field that registration methods focus on. I would suggest adding different descriptors on both 3DMatch and 3DLoMatch, those descriptors should cover a wide range of outlier rates. (as on synthetic data, the test covering different outlier rates is included, but on real data, there is no such setting) Moreover, using FCGF descriptors on 3DMatch but Predator on 3DLoMatch is weird. It looks like that FCGF doesn't perform well on 3DLoMatch, which leads to bad results for the proposed method. The authors have to choose another descriptors for 3DLoMatch.

3. More  visualization results on indoor and outdoor scenes are encouraged. And it is also better to directly visualize the correspondences before and after using the proposed method.

4. Some closely related works may be missing:

  [1]. Yu et al. CoFiNet: Reliable Coarse-to-fine Correspondences for Robust Point Cloud Registration, NeurIPS 2021;

  [2]. Ao et al. SpinNet: Learning a General Surface Descriptor for 3D Point Cloud Registration, CVPR 2021

   [3]. Qin et al. Geometric Transformer for Fast and Robust Point Cloud Registration, CVPR 2022;

   [4]. Want et al. You Only Hypothesize Once':' Point Cloud Registration with Rotation-equivariant Descriptors, ACM MM, 2022;

   [5]. Yu et al. RIGA: Rotation-Invariant and Globally-Aware Descriptors for Point Cloud Registration, TPAMI 2024
   [6]. Yu et al. Rotation-Invariant Transformer for Point Cloud Matching, CVPR 2023;

**Questions:**

See weaknesses.

---

> ### Author Response · Authors · 2024-11-23
> **Response to Reviewer k1rY**
>
> We greatly appreciate your positive feedback and constructive suggestions on our paper. We are glad that you found our approach innovative. Below, we respond to your specific comments:
>
> > W1: In the experimental results, it seems there is no time analysis. It is better to compare the proposed method to existing ones in terms of time cost;
> - We appreciate your suggestions. Regarding the analysis of time cost, we also understand that this is an important consideration. As you said, in this study, we propose a novel method that can significantly improve registration accuracy without additional training procedures. Our main focus is to verify the accuracy and robustness of the proposed method in handling different outliers, so we prioritize these metrics in the experimental section. Given our research objectives, we choose to focus on the RE, TE, RR performance of the algorithm. In addition, we have modified Table 1 to provide the runtime and computational overhead information required by Hessian, which will provide a complete performance picture. Thank you for your understanding and support.
>
> > W2: The experiments on Indoor Scenes may be insufficient, as this is the main field that registration methods focus on. I would suggest adding different descriptors on both 3DMatch and 3DLoMatch, those descriptors should cover a wide range of outlier rates. (as on synthetic data, the test covering different outlier rates is included, but on real data, there is no such setting) Moreover, using FCGF descriptors on 3DMatch but Predator on 3DLoMatch is weird. It looks like that FCGF doesn't perform well on 3DLoMatch, which leads to bad results for the proposed method. The authors have to choose another descriptors for 3DLoMatch.
>
> - We fully agree that it is important to add different descriptors in indoor scene experiments. In the preliminary experimental design, a single descriptor was selected for 3DMatch and 3DLoMatch, mainly based on the usage frequency of these descriptors. We have added experiments with two other descriptors regarding SC2-PCR and MAC. Specifically, 3DMatch uses both FCGF and FPFH. 3DLoMatch uses both FCGF and Predator (**Tables 3 and 4 in revised manuscript**). We are pleased to find that our method can still achieve better results among different descriptors. We believe that these supplementary experiments will make our paper more comprehensive.
>
> > W3: More visualization results on indoor and outdoor scenes are encouraged. And it is also better to directly visualize the correspondences before and after using the proposed method.
>
> - We agree that more visualization results will help to more intuitively demonstrate the performance of our method. We have added more visualization results for indoor and outdoor scenes (**Figures 3 and 4 in supplementary material**). We believe that these supplementary materials will further enhance the quality and persuasiveness of the paper. Thanks again for your suggestions.
>
> W4: Some closely related works may be missing:
>
> [1]. Yu et al. CoFiNet: Reliable Coarse-to-fine Correspondences for Robust Point Cloud Registration, NeurIPS 2021;
>
> [2]. Ao et al. SpinNet: Learning a General Surface Descriptor for 3D Point Cloud Registration, CVPR 2021
>
> [3]. Qin et al. Geometric Transformer for Fast and Robust Point Cloud Registration, CVPR 2022;
>
> [4]. Want et al. You Only Hypothesize Once':' Point Cloud Registration with Rotation-equivariant Descriptors, ACM MM, 2022;
>
> [5]. Yu et al. RIGA: Rotation-Invariant and Globally-Aware Descriptors for Point Cloud Registration, TPAMI 2024
>
> [6]. Yu et al. Rotation-Invariant Transformer for Point Cloud Matching, CVPR 2023;
>
> - Thank you for your thoughtful suggestions. These related works provide valuable references for this paper and have important implications and guiding significance for future research. We have added these related works in the revised version. (Yu et al., 2021), (Ao et al, 2021), (Qin et al, 2022), (Wang et al, 2022), (Yu et al, 2024), (Yu et al, 2023), (Qin et al, 2023)
>
> Finally, we would like to express our gratitude again for your time and effort in reviewing our paper. Please do not hesitate to let us know if you have any further concerns or comments. We would be happy to address them.

---

> ### Author Response · Authors · 2024-11-26
> **Official Comment by Authors**
>
> Dear Reviewer k1rY,
>
> We sincerely appreciate your time and effort in reviewing our paper and providing valuable suggestions.
>
> Since there will be no second stage of author-reviewer discussions, and the discussion phase is drawing to a close, we would to confirm whether our responses have effectively addressed your concerns. If you require further clarification or have additional concerns, please don’t hesitate to contact us. We are more than willing to continue our communication with you.
>
> Best regards.
>
> The Authors

---

> ### Comment · Reviewer_k1rY · 2024-11-27
>
> I think the authors addressed most of my concerns well. I will keep my positive score. However, I still suggest the authors adding more time analysis for the overall comparisons with existing methods, not just only showing the running time of some designed blocks with out references.

---

> ### Author Response · Authors · 2024-11-27
> **Official Comment by Authors**
>
> Thank you for your thoughtful feedback and for taking the time to assess our revisions. We sincerely appreciate your recognition of the paper and your positive comments. We would add time analysis in the final manuscript to further improve our paper.

---

### Official Review · Reviewer_dmuo · 2024-11-01

**Soundness:** 3
**Presentation:** 3
**Contribution:** 3
**Rating:** 6
**Confidence:** 3

**Summary:**

This paper introduces the Adaptive Graduated Non-Convexity (AGNC) method for point cloud registration, addressing challenges related to convergence to local minima caused by incorrect correspondences. By utilizing the Hessian of the cost function, AGNC adaptively adjusts the scale of the optimization landscape, enhancing  robustness without relying on a fixed optimization schedule, which can be costly or less effective. Experimental results demonstrate that AGNC significantly outperforms existing techniques on both simulated and real datasets, achieving reliable registration even with up to 99% outlier rates.

**Strengths:**

Overall, the paper is clearly written, excluding the equations, which could benefit from additional clarification. The algorithm is easy to follow, and the idea is original, effectively addressing the challenges of fixed scaling of cost function posed by graduated non-convexity methods.

**Weaknesses:**

This paper introduces an adaptive method for adjusting the scale of the cost function in graduated non-convexity (GNC) to address challenges related to convergence steps. However, there are several areas for improvement:

1-While the paper presents experimental validation using a toy example to illustrate the effects of the adaptive scheme on convergence iterations, it lacks critical information on runtime and the computational overhead required for Hessian. Including runtime data in Table 1 would be beneficial, as it would provide a complete picture of performance by accounting for the overhead of Hessian computation and binary search.

2-The paper contains many acronyms, such as FGCF and LSQ, without definitions. It is important to clarify these terms early on to enhance reader understanding. Additionally, several notations, like "r" in Equation 1, are introduced without explanation. These should be defined as soon as they are presented. Equations 6 through 11 also contain unexplained notations, which can confuse readers.

3-The assertion that “…. H in Eq. 2 obtained at µ_{k+1} is ensured to remain positive definite, then the new estimate z_{k+1} is guaranteed to be in the same convergence domain as the previous iteration” requires citation or proof. Providing a reference would strengthen the validity of this claim.

4-An insightful addition would be to include experiments assessing the impact of a multi-task knowledge-sharing mechanism with fixed scaling in Table 6, alongside the number of iterations and runtime required for convergence. Currently, the errors without knowledge sharing are worse than those of the fixed scaling scheme, raising questions about whether the adaptive scale adjustment primarily enhances convergence speed rather than accuracy. Adding columns to compare knowledge sharing in fixed scaling settings and the number of convergence stages would clarify the contributions of each component.

5-I recommend visualizing the outputs from competitive methods in Figure 4 and using bright colours to enhance clarity in the visualizations (minor)

6-In Tables 3 and 4, it is unclear if "ER" and "Et" refer to rotation error and translation error, respectively. If so, for consistency with the previous definitions, they should be labelled as "RE" and "TE" as mentioned in line 344.

7-I am not convinced of the accurate performance of this method, as the criteria for successful registration (line 431) in Tables 3 and 4 seem too lenient for both indoor and object-level settings. Similar concerns also arise for Table 5.

Please correct the spelling of "black-Rangarajan" duality. It would also be beneficial to include a brief sentence explaining what this duality is about for the reader’s clarity.
In line 305, please change the reference from "(line 4)" to "(line 5)"

**Questions:**

1-Given that computing the Hessian is typically expensive, could the authors please discuss whether alternative routines, such as Adam or RMSprop, could serve as equivalent methods for scaling? How would these compare in terms of performance and convergence speed?
2-The paper states that "little attention has been paid to how the scaling factor is determined." Please provide references to existing techniques that address this issue. Including comparisons to these methods would enhance the context of this work.
3- Once the Hessian is computed, how valuable is its inclusion in the cost function? I would expect this to lead to faster convergence with minimal overhead. Can authors provide insights on this aspect?
4-How valid is it to consider different optimization stages as different tasks? Could authors elaborate on this concept and its implications for the proposed method?
5-How is the final value of \mu_{final}determined?
6-Could authors add extra columns in Tables 3 and 4 to indicate the range of initial total translation and rotation offsets? The translation errors of greater than 6 cm and 9 cm for even the best performances seem unacceptable for practical applications in small indoor settings.

---

> ### Author Response · Authors · 2024-11-23
> **Response to Reviewer dmuo (1/3)**
>
> Thank you for taking the time to review our work. We have carefully addressed the questions and concerns raised, and have made updates to the manuscript to improve clarity and understanding. We will address each point below.
>
> > W1: While the paper presents experimental validation using a toy example to illustrate the effects of the adaptive scheme on convergence iterations, it lacks critical information on runtime and the computational overhead required for Hessian. Including runtime data in Table 1 would be beneficial, as it would provide a complete picture of performance by accounting for the overhead of Hessian computation and binary search.
>
> - We completely agree with you that information on runtime and the computational overhead required for Hessian would provide a complete picture of performance. We have added two columns to Table 1 specifically to show the runtime data under different experimental settings.
>
>  **Table 1: Comparison of different updating methods of $\zeta$.**
>
>   | GNC strategy          | Stages | Runtime | Hessiantime | Accuracy |
> |-----------------------|--------|---------|-------------|----------|
> | Fixed small $\zeta$   | 16     | 4.98    | -           | Medium   |
> | Fixed large $\zeta$   | 6      | 2.02    | -           | Low      |
> | Ours adaptive $\zeta$ | 8      | 3.61    | 1.08        | High     |
>
>
> - These results show that although there is a certain overhead in Hessian computation and binary search, the overall runtime is still within an acceptable range with better performance. We will include these changes in the revised version and ensure that our study can provide readers with deeper insights.
>
> > W2: The paper contains many acronyms, such as FGCF and LSQ, without definitions. It is important to clarify these terms early on to enhance reader understanding. Additionally, several notations, like "r" in Equation 1, are introduced without explanation. These should be defined as soon as they are presented. Equations 6 through 11 also contain unexplained notations, which can confuse readers.
>
> - We have provided clear definitions for all acronyms in the revised manuscript and explained the meaning of each symbol immediately upon its first appearance. This enhances the readability and understanding of the paper.
>
> > W3: The assertion that “…. H in Eq. 2 obtained at µ_{k+1} is ensured to remain positive definite, then the new estimate z_{k+1} is guaranteed to be in the same convergence domain as the previous iteration” requires citation or proof. Providing a reference would strengthen the validity of this claim.
>
> - In the {k+1} iteration, if we update the scale to µ_{k+1} ensuring that the Hessian remains positive definite, then we have likely ensured that the new z_{k+1} is in the same convergence domain as the previous iteration. This is true since we initialize optimization for the {k+1} iteration at z_{k} and converge to the local minimum by IRLS. We have cited relevant references in the paper to support this assertion (Andrew&Gao, 2007; Kohetal., 2007; Ochsetal., 2013).
>
> > W4: An insightful addition would be to include experiments assessing the impact of a multi-task knowledge-sharing mechanism with fixed scaling in Table 6, alongside the number of iterations and runtime required for convergence. Currently, the errors without knowledge sharing are worse than those of the fixed scaling scheme, raising questions about whether the adaptive scale adjustment primarily enhances convergence speed rather than accuracy. Adding columns to compare knowledge sharing in fixed scaling settings and the number of convergence stages would clarify the contributions of each component.
>
> - We understand your concern about whether the adaptive scale adjustment primarily improves convergence speed rather than accuracy. We have added columns in Table 6 to compare knowledge sharing in fixed scaling settings and the number of convergence stages to clarify the contribution of each component. As shown in **Table 6 in revised manuscript**, the adaptive scale adjustment reduces the number of stages by about half, while effectively reducing both the rotation error and the translation error. As you said, the errors without knowledge sharing are worse. The reason is that tracking the local minimum sometimes leads to a solution far away from the ground truth. The adaptive scale adjustment of graduated non-convexity effectively not only effectively improves the registration speed, but also improves the registration accuracy. The knowledge sharing strategy overcomes the severe failure conditions caused by high outliers to jump out of the local minimum.

---

> > ### Comment · Reviewer_dmuo · 2024-11-26
> > **Clarification of role of terms in the equations and convergence steps**
> >
> > Thank you for your efforts in addressing my feedback. I have a few additional comments. The text still lacks specific definitions or detailed explanations of the roles of many terms, such as the gradients \( g_{\text{LSQ},i} \) (the exact meaning of each term is unclear in equation 7), as well as \( l_i \) and \( m_i \), which are important for understanding their roles in the registration process. Additionally, the notations \( z_r \) and \( z_s \) require further clarification—are these partial derivatives with respect to rotation and translation, or something else?
> >
> > Also, runtime is missing in Table 6. The results with knowledge sharing for fixed scheme are much better than those without knowledge sharing but the convergence steps remain the same, which raises a concern. Why did the convergence steps not decrease for the fixed scheme with knowledge sharing despite achieving relatively high accuracy? Could you please clarify the criteria for convergence in this case?

---

> > > ### Author Response · Authors · 2024-11-27
> > > **Response to Reviewer dmuo**
> > >
> > > Thank you for your continued attention and valuable feedback on our paper.
> > > > C1: Thank you for your efforts in addressing my feedback. I have a few additional comments. The text still lacks specific definitions or detailed explanations of the roles of many terms, such as the gradients ( g_{\text{LSQ},i} ) (the exact meaning of each term is unclear in equation 7), as well as ( l_i ) and ( m_i ), which are important for understanding their roles in the registration process. Additionally, the notations ( z_r ) and ( z_s ) require further clarification—are these partial derivatives with respect to rotation and translation, or something else?
> > > - Thank you for your valuable feedback. We have made detailed additions and clarifications in the revised manuscript regarding the definitions and explanations of terms. Specifically, g_{\text{LSQ},i} is the gradient of the least squares cost at the i-th residual. H_{\text{LSQ},i} is the Hessian of the least squares cost at the i-th residual. l_i and m_i are factors for weight adjustment based on the residuals, which are used to modify the gradient and Hessian. They reflect the sensitivity of the loss function to the residual and help balance the treatment of outliers during point cloud registration process. In equation 7, x_i is the coordinate of the i-th point in the source point cloud. R is the rotation matrix. r_i is the residual of the i-th correspondence. In equation 5, the partial derivatives of z_r and z_s are with respect to the two components of rotation and translation.
> > > - We have updated the relevant content in the revised manuscript to ensure that readers can intuitively understand the key roles of these symbols in our method.
> > >
> > > > C2: Also, runtime is missing in Table 6. The results with knowledge sharing for fixed scheme are much better than those without knowledge sharing but the convergence steps remain the same, which raises a concern. Why did the convergence steps not decrease for the fixed scheme with knowledge sharing despite achieving relatively high accuracy? Could you please clarify the criteria for convergence in this case?
> > > - Thank you for your feedback. We have supplemented the runtime. (**in Table 6 in revised manuscript**)
> > > - The purpose of the knowledge sharing mechanism is to achieve collaborative optimization of non-convex cost functions at different levels and help the algorithm escape from the local optimal solution. Therefore, the results with knowledge sharing overcome the serious failure to fall into the local minimum (this failure may lead to a solution far from the true value). Compared with the results without knowledge sharing, the registration accuracy is improved. In other words, knowledge sharing reduces the errors caused by some solutions far from the true value.
> > > - The convergence criterion of the fixed scheme is set so that when the registration error (i.e., the Euclidean distance from the transformed point of the source point cloud to the target point cloud) changes very small between two iterations, the registration is considered to have converged. Because in point cloud registration, when the registration error is very small and no longer significantly improved, the registration process can be considered complete. The number of convergence steps has no significant relationship with the knowledge sharing mechanism. Under the fixed scheme, the algorithm with or without knowledge sharing will converge near a local optimal solution. In this case, the number of convergence steps is mainly determined by the magnitude of the error change. Therefore, even if knowledge sharing improves registration accuracy, the convergence step may remain the same.
> > > - We have completed the relevant revisions in the revised manuscript. Thank you again for your valuable comments that have spurred our further thinking.

---

> > > > ### Comment · Reviewer_dmuo · 2024-11-27
> > > >
> > > > Thank you for updating the text. Just a few final remarks:
> > > >
> > > > Equation 7 includes square brackets with some operators outside of them. Could you please clarify and include a description of this in the main text for better understanding?
> > > >
> > > >  It would be helpful to include specific pointers in the main text that guide the reader to relevant sections in the supplementary material.
> > > >
> > > > Please ensure that all units are included such as with the runtime and others if missing.

---

> > > > > ### Author Response · Authors · 2024-11-28
> > > > > **Response to Reviewer dmuo**
> > > > >
> > > > > Thank you for your continued attention and thoughtful feedback.
> > > > > > Comment: Thank you for updating the text. Just a few final remarks: Equation 7 includes square brackets with some operators outside of them. Could you please clarify and include a description of this in the main text for better understanding? It would be helpful to include specific pointers in the main text that guide the reader to relevant sections in the supplementary material. Please ensure that all units are included such as with the runtime and others if missing.
> > > > > - Thank you for your careful review. Regarding the issue of the operators outside the square brackets in Equation 7, we have clarified and supplemented the detailed description in the revised manuscript to ensure that readers can better understand its meaning. "\left[~\right]_{\times} is a skew-symmetric matrix operation, which is used to convert a coordinate vector into a skew-symmetric matrix form for cross-multiplication with the rotation vector. {\top} is the transpose of the matrix."
> > > > > - We have added specific pointers in the revised manuscript to help readers quickly locate relevant content in the supplementary materials.
> > > > > - Thanks for your suggestions. We have carefully checked and supplemented all required units (such as runtime) to ensure that every data in the paper has complete unit annotations.
> > > > >
> > > > > Thank you again for your detailed feedback, which helped us further improve the expression and content of the paper. All revisions have been completed in the revised manuscript.

---

> > > > > ### Author Response · Authors · 2024-11-28
> > > > > **Thank you for your continued attention and valuable feedback**
> > > > >
> > > > > We would like to express our gratitude again for your time and effort in reviewing our paper. Your insights not only improved the quality and comprehensiveness of our paper but also greatly deepened our thinking. Please do not hesitate to let us know if you have any further concerns or comments. We would be happy to address them.

---

> ### Author Response · Authors · 2024-11-23
> **Response to Reviewer dmuo (2/3)**
>
> > W5: I recommend visualizing the outputs from competitive methods in Figure 4 and using bright colors to enhance clarity in the visualizations (minor).
>
> - Thank you for your suggestion regarding the visualization of the figure. We have redrawn Figure 4 with bright colors. More visualizations of the registration results with competitive methods are provided in the **Figure 1 of supplementary material**.
>
> > W6: In Tables 3 and 4, it is unclear if "ER" and "Et" refer to rotation error and translation error, respectively. If so, for consistency with the previous definitions, they should be labeled as "RE" and "TE" as mentioned in line 344.
>
> - Thank you for your careful reading. We have corrected ER and Et to RE and TE in the revised manuscript to maintain consistency with the previous definitions. At the same time, we have checked and ensured the accuracy and consistency of all terms in the manuscript.
>
> > W7: I am not convinced of the accurate performance of this method, as the criteria for successful registration (line 431) in Tables 3 and 4 seem too lenient for both indoor and object-level settings. Similar concerns also arise for Table 5.
>
> - Thank you for your attention to the performance evaluation criteria of this method. We understand that you have doubts about the strictness of the successful registration criteria in Tables 3, 4, and 5. These criteria are not set arbitrarily, but refer to the evaluation criteria in the comparative methods [1, 2, 3] (Zhang et al, 2023; Chen et al, 2022b; Huang et al, 2024). These references provide us with recognized evaluation indicators and threshold settings to ensure the scientificity and rationality of our experiments.
> - Regarding the issue that the threshold setting may appear too loose, it is because the point cloud registration problem on the 3DMatch, 3DLoMatch, and KITTI datasets is very challenging. First, the point cloud overlap rate in these datasets is generally low, which means that fewer feature pairs can be used for registration, which increases the difficulty of registration. Second, the noise and occlusion contained in the point cloud data will interfere with the feature extraction and matching process, increasing false matches. For more information on the challenges of the dataset, please see **Table 1 in supplementary material**. We have cited the above references in the revised manuscript and further explained the reasons for our selection of these criteria.
>
> References
>
> [1] 3D Registration with Maximal Cliques, CVPR, 2023
>
> [2] SC^2-PCR: A Second Order Spatial Compatibility for Efficient and Robust Point Cloud Registration, CVPR, 2022
>
> [3] Efficient and Robust Point Cloud Registration via Heuristics-guided Parameter Search, TPAMI, 2024
>
> > W8: Please correct the spelling of "black-Rangarajan" duality. It would also be beneficial to include a brief sentence explaining what this duality is about for the reader’s clarity. In line 305, please change the reference from "(line 4)" to "(line 5)"
>
> - We appreciate your careful review and valuable suggestions. We have corrected it to "black-Rangarajan" duality and then added a brief sentence with reference. "This duality provides a way to convert traditional line process methods and robust statistical methods into each other (Black & Rangarajan, 1996)." In addition, we have also corrected the reference in line 305 from "(line 4)" to "(line 5)".
>
> > Q1: Given that computing the Hessian is typically expensive, could the authors please discuss whether alternative routines, such as Adam or RMSprop, could serve as equivalent methods for scaling? How would these compare in terms of performance and convergence speed?
>
> - We recognize that calculating the Hessian matrix does have the problem of high computational cost. Regarding whether optimization algorithms such as Adam or RMSprop can be used instead of Hessian for scale adjustment, we believe that although these two algorithms perform well in many tasks, their core mechanisms are different from the goals of our method. Adam and RMSprop mainly adjust the learning rate by adaptively estimating the first moment and second moment [1, 2]. However, they do not directly utilize second-order information and differ from the Hessian-based methods in terms of convergence speed and final performance. Adam and RMSprop do not explicitly consider the structural information of the Hessian, so they may not be able to effectively capture the global optimal solution when facing the point cloud registration optimization problem with strong non-convexity. Regarding your suggestion of using the Adam and RMSprop algorithms as alternatives, we believe that this is a research direction worthy of further exploration in the point cloud registration task.
>
>   References
>
>   [1] Adam: A method for stochastic optimization, 2014
>
>   [2] An overview of gradient descent optimization algorithms, 2016

---

> ### Author Response · Authors · 2024-11-23
> **Response to Reviewer dmuo (3/3)**
>
> > Q2: The paper states that "little attention has been paid to how the scaling factor is determined." Please provide references to existing techniques that address this issue. Including comparisons to these methods would enhance the context of this work.
>
> - Thank you for pointing out the need for references. In the literature, some methods explore the issue of choosing a scaling factor. We have included these references (Hazan et al., 2016; Le & Zach, 2020) in the revised manuscript.
>
> > Q3: Once the Hessian is computed, how valuable is its inclusion in the cost function? I would expect this to lead to faster convergence with minimal overhead. Can authors provide insights on this aspect?
>
> - Thank you for your comments. Indeed, the calculation of the Hessian matrix will bring a certain amount of time overhead. In our method, the introduction of the Hessian matrix is mainly to provide more accurate second-order derivative information, which helps the algorithm to more effectively adjust the step size when approaching the optimal solution, thereby achieving faster convergence. We have added two columns to **Table 1 in revised manuscript** specifically to show the time cost under different experimental settings. These results show that although there is some overhead in the Hessian computation, better convergence is achieved with less overhead.
>
> - In the future, we believe it would be possible to consider optimizations (e.g., using approximation methods) to try to reduce this overhead further.
>
> > Q4: How valid is it to consider different optimization stages as different tasks? Could authors elaborate on this concept and its implications for the proposed method?
>
> - Thank you for your attention to the concept of different optimization stages. It is an innovative idea to treat different optimization stages as different tasks, which is based on the observation that the shape of the cost function or the optimal solution has certain similarities at various stages of the optimization process. This approach is inspired by multi-task learning, where different but related tasks can improve the optimization performance by sharing knowledge.
>
> - In this paper, we achieve the collaborative optimization of non-convex cost functions at different levels through a multi-task knowledge sharing mechanism, which helps to improve the success rate of point cloud registration under the challenging condition of a high outlier rate. The ablation experiments in Table 6 further verify the effectiveness of the strategy. We find that without knowledge sharing, the registration results are very poor. The reason is that the point cloud registration method is at risk of settling in the local minimum under high outliers, which will lead the results far away from the true solution.
>
> > Q5: How is the final value of \mu_{final}determined?
>
> - Thank you for pointing this out. The choice of \mu_{final} was based on the results of experimental analysis. We experimentally evaluated the impact of different \mu_{final} values on the point cloud registration performance and selected the value that achieved the best registration accuracy and robustness on multiple datasets. In addition, we considered computational efficiency to ensure that the selected \mu_{final} value can complete the optimization process in a reasonable time.
>
> > Q6: Could authors add extra columns in Tables 3 and 4 to indicate the range of initial total translation and rotation offsets? The translation errors of greater than 6 cm and 9 cm for even the best performances seem unacceptable for practical applications in small indoor settings.
>
> - Thank you for your suggestion. We have listed the contents of all datasets in detail in **Table 1 of  supplementary material**. We also provide a further introduction to the datasets. We agree that in some precise indoor environment application scenarios, such as industrial parts inspection. Large translation errors and rotation errors are unacceptable. However, the point cloud registration problem on the experimental dataset in this paper is very challenging. Point clouds have different overlap rates, noise, occlusion, and other interferences. These factors reduce the number of valid feature pairs and increase false matches. The registration performance of all methods in small indoor environments is the average result on this dataset. Not all models exceed the translation error of 6 cm and 9 cm.
>
> Finally, we would like to express our gratitude again for your time and effort in reviewing our paper. Please do not hesitate to let us know if you have any further concerns or comments. We would be happy to address them.

---

> > ### Comment · Reviewer_dmuo · 2024-11-26
> >
> > Thank you for your response and for carefully addressing the comments. However, considering the presence of other (adaptive) schemes in the literature, I feel the paper would be more complete with a comparison to these methods. It would be helpful to include comparisons in terms of convergence steps, runtime, and accuracy to provide a more complete context for the work.
> >
> >  While I understand that these thresholds have been adopted from the literature, I would appreciate it if you could provide the initial translation and rotation offset ranges in the tables as this would give a deeper understanding of the scale of errors (Table 1 in the supplementary material does not contain this information). Although these criteria are taken from literature, I find them somewhat lenient. Given that these are considered successful registrations, I would expect that applying a refined registration method to these results would yield the correct registration with minimal errors. If this is not the case, it would be helpful to include a discussion in the text regarding the specific applications that can tolerate up to 30 cm of translation error indoors, and similarly for outdoor settings. This would provide a better understanding of the practical implications of these thresholds.

---

> > > ### Comment · Reviewer_dmuo · 2024-11-26
> > >
> > > If the introduction includes a motivation for performing registration between two point clouds with 99% outliers, along with examples of scenarios where this issue arises, it would provide valuable context for the reader.

---

> > > > ### Author Response · Authors · 2024-11-27
> > > > **Response to Reviewer dmuo**
> > > >
> > > > Thank you for your continued attention and valuable feedback on our paper.
> > > > > C1: If the introduction includes a motivation for performing registration between two point clouds with 99% outliers, along with examples of scenarios where this issue arises, it would provide valuable context for the reader.
> > > > - Thank you for your valuable comments. We fully agree with the suggestion to add motivation and real-world examples of 99% outlier point cloud registration in the Introduction.
> > > > - In the revised paper (**lines 41-44**), we elaborate on the motivation for point cloud registration with high outlier rates. "High outlier rates (sometimes exceeding 99%) are a typical feature of point cloud keypoint detection and registration, which poses a great challenge to point cloud registration (Huang et al, 2020; Qin et al, 2022; Yuan et al, 2023). This challenge is common, where matching often produces false correspondences due to noise, occlusions, and sensor errors."
> > > > - We also further introduce an example of an autonomous driving scenario (**lines 44-47**) that often exhibits high outlier rates. "For example, in autonomous driving, LiDAR scanning is often interfered with by dynamic objects such as cars and pedestrians and contains a lot of background noise (Bogdoll et al, 2022). Registration methods must effectively handle these outliers to ensure proper functioning of safety systems."
> > > > - By adding these contents, we hope to further enhance the readers' understanding of the motivation of the problem and emphasize the practical application value. Thank you again for your valuable comments, we have implemented these improvements in the introduction of the revised manuscript.

---

> ### Author Response · Authors · 2024-11-26
> **Official Comment by Authors**
>
> Dear Reviewer dmuo,
>
> We sincerely appreciate your time and effort in reviewing our paper and providing valuable suggestions.
>
> We understand that you may be extremely busy at this time. But we would to confirm whether our responses have effectively addressed your concerns. We hope that you could kindly update the rating if your questions have been addressed. We are also happy to answer any additional questions before the rebuttal ends.
>
> Best regards,
>
> The Authors

---

> ### Author Response · Authors · 2024-11-27
> **Response to Reviewer dmuo**
>
> Thank you for your continued attention and valuable feedback on our paper.
> > C1: Thank you for your response and for carefully addressing the comments. However, considering the presence of other (adaptive) schemes in the literature, I feel the paper would be more complete with a comparison to these methods. It would be helpful to include comparisons in terms of convergence steps, runtime, and accuracy to provide a more complete context for the work.
> - Thank you for your feedback and suggestions. We have included a comparison with existing scale adaptive schemes in the supplementary material (**in Table 2 in supplementary material**). Our method AGNC is compared with two different scaling schemes, GradOpt (Hazan et al., 2016) and ASKER(Le & Zach, 2020). Table 2 reports the results of convergence steps, runtime, and relative accuracy. Compared with the second-ranked ASKER method, the convergence steps are reduced by 11 stages, the runtime is reduced by 3.92 and with higher accuracy. The results show that our AGNC achieves the best results.
> - These additions more fully demonstrate the advantages of our approach. Thank you very much for your suggestions, and we have added these improvements to the supplementary materials.
>
> > C2: While I understand that these thresholds have been adopted from the literature, I would appreciate it if you could provide the initial translation and rotation offset ranges in the tables as this would give a deeper understanding of the scale of errors (Table 1 in the supplementary material does not contain this information). Although these criteria are taken from literature, I find them somewhat lenient. Given that these are considered successful registrations, I would expect that applying a refined registration method to these results would yield the correct registration with minimal errors. If this is not the case, it would be helpful to include a discussion in the text regarding the specific applications that can tolerate up to 30 cm of translation error indoors, and similarly for outdoor settings. This would provide a better understanding of the practical implications of these thresholds.
> - Thank you for your careful reading and valuable comments. We agree that counting the initial translation and rotation offset ranges helps readers understand the scale of the errors. We have added columns to the table to explicitly show the initial translation and rotation offset ranges (**in Table 1 in supplementary material**). It can be observed that in addition to the interference of noise, outliers, and limited overlap, the initial rotation and translation errors are also very large.
> - We understand your concern about the evaluation criteria. The point cloud registration problem in the datasets used in the experiments is very challenging. These datasets contain noise, outliers, and sensor differences, which make the registration task very difficult. Therefore, although the evaluation criteria seem loose, they reflect the actual difficulty of performing registration under these conditions. All compared methods include the refined method in their papers to minimize the error. Despite these optimizations, it is still very difficult to further reduce the registration error. Under idealized conditions, without interference such as outliers or noise, point cloud registration can achieve very accurate results. However, in the real world, registration is still a complex task, especially when considering the imperfections of the data.
> - Based on your suggestion, we have also added and discussed two practical application examples. (**in Section 4 in supplementary material**) One is a virtual reality scenario, and the other is an autonomous driving scenario. Translation errors within the tolerable range are acceptable for both.
> - We agree with the reviewer that further reducing the error is essential to improve the applicability of the registration. This will be the core goal of our future research. We are very grateful for your suggestions, which will help us better demonstrate the practical significance and application value of our work. We have fully considered and implemented these improvements in the revised manuscript.

---

> ### Comment · Reviewer_dmuo · 2024-11-27
>
> Thank you for addressing the review. While I am not certain if a 30 cm offset is still considered acceptable in virtual reality, I do believe that a 60 cm threshold (for Table 5) is likely too large to be considered within a tolerable range. In the context of environmental modeling for autonomous driving, I imagine that such a significant error could lead to critical localization issues that may lead to other problems—such as vehicles drift off the road, failing to follow lanes (without land detection in place), or even colliding with pre-modeled objects in the map, like traffic lights. Please feel free to correct me if my understanding here is incorrect.

---

> ### Author Response · Authors · 2024-11-28
> **Response to Reviewer dmuo**
>
> > Comment: Thank you for addressing the review. While I am not certain if a 30 cm offset is still considered acceptable in virtual reality, I do believe that a 60 cm threshold (for Table 5) is likely too large to be considered within a tolerable range. In the context of environmental modeling for autonomous driving, I imagine that such a significant error could lead to critical localization issues that may lead to other problems—such as vehicles drift off the road, failing to follow lanes (without land detection in place), or even colliding with pre-modeled objects in the map, like traffic lights. Please feel free to correct me if my understanding here is incorrect.
> - Thank you for your in-depth review and valuable comments of our paper. We completely agree with your perspective that in high-precision application scenarios within autonomous driving, such as precise lane keeping, even minor errors can have serious consequences. Your concern regarding the 60 cm threshold is entirely justified, as it could indeed impact the positioning accuracy and safety of autonomous vehicles.
> - In practice, the autonomous driving system does not solely rely on point cloud registration results but continuously corrects and optimizes vehicle positioning through real-time data updates and advanced multi-sensor registration and fusion technologies. These technologies work in concert to provide continuous, high-precision positional information. Therefore, even if there are certain errors in point cloud registration, these errors may be corrected by subsequent sensor data fusion processes to ensure the safe operation of the vehicle. Correct positioning and path planning can also be maintained in the face of lane changes. Of course, we also recognize that lower registration errors will further enhance the safety of autonomous driving.
> - To further validate the performance of our AGNC method under more stringent error requirements, we have adjusted the threshold and tested the performance of AGNC under a translation error threshold of 30 cm. Compared to the original threshold of 60cm, where the AGNC method achieved an average registration success rate of 99.12%, under the updated threshold of 30cm, the AGNC method achieved an average registration success rate of 87.26%. The results demonstrate that our method still exhibits a high registration success rate under more stringent conditions.
> - We believe that these supplementary results can better respond to your concerns about error tolerance. Thank you again for your feedback. We have improved the discussion section of the supplementary materials.

---

> ### Author Response · Authors · 2024-11-30
> **Looking forward to your feedback**
>
> Dear reviewer dmuo,
>
> Thank you once again for your valuable insights. As the discussion period is coming to an end, we would like to confirm whether we have adequately addressed your concerns. If you have any additional comments or concerns, we would be happy to address them during the remainder of the discussion period. We have noticed that you have improved your score. We sincerely appreciate your recognition and the score adjustment.
>
> Sincerely,
>
> Authors

---

> > ### Comment · Reviewer_dmuo · 2024-12-01
> >
> > Thank you so much for your valuable efforts in adequately addressing the comments.
> >
> > However, I remain somewhat unconvinced that the application cases included in the supplementary material—specifically those that accept a 30 cm indoor and 60 cm outdoor registration error—are entirely accurate. I believe it would be more appropriate to remove these examples from the supplementary material to prevent any potential misguidance, unless we can support them with relevant references from the literature.

---

> > > ### Author Response · Authors · 2024-12-02
> > > **Response to Reviewer dmuo**
> > >
> > > Thank you for your valuable feedback and for taking the time to assess our revisions again. We will remove the inappropriate examples in the final version to prevent any potential misguidance. We sincerely appreciate your recognition and in-depth review of the paper.

---

### Official Review · Reviewer_aCKU · 2024-11-02

**Soundness:** 3
**Presentation:** 2
**Contribution:** 3
**Rating:** 8
**Confidence:** 4

**Summary:**

This paper proposed a robust point cloud registration method based on Adaptive Gradient non-convexity (AGNC). By monitoring the correct definiteness of the Hessian of the cost function, the scale of gradient non-convexity is adaptively reduced without the need for a fixed optimization schedule. AGNC is far superior to existing methods in terms of both robustness and accuracy, achieving accurate registration results even at extreme 99% outlier rates.

**Strengths:**

1. Although the idea of this paper is simple, and it is very important for the field of point cloud registration, and I think this paper is very meaningful. Simple is best.

2. For the inherent problem of point cloud registration: high outlier rate, it provides a feasible idea: MULTI-TASK KNOWLEDGE SHARING, which can also perform well in high outlier rate.

**Weaknesses:**

1.The experimental setup needs to be improved.
 I have my doubts about the experimental setup. A major contribution of this paper is that high outlier rates can also perform well. As for the evaluation of high outlier rate, only on the simulation data set, however, as shown in FIG. 4, the current methods can achieve good registration. This does not speak to the strengths of your approach. On the contrary, on 3DMatch dataset, with high outlier rate, it is difficult for existing methods to achieve robust registration. It is suggested to add the experimental results of Outlier rate=99% in 3DMatch dataset, which can make this paper more convincing. "Accurate registration results are obtained even at the extreme 99% outlier rate." "Is a bit of an exaggeration.

2.Some  typos.
What do we mean by ER and Et in Tables 3 and 4 and 5? Should it be RE and TE?

**Questions:**

1.For the experiments on 3DMatch/3DLoMatch dataset to be registered with only one feature, I think it is not enough, please refer to the experimental setup of SC2-PCR and MAC. In addition, 3DMatch uses FCGF and 3DLoMatch uses Predator for feature extraction, which is kind of intentional to show the reader better results.

2. It is suggested that ablation experiments should also be done on the 3DMatch data set. Only the contribution to accuracy can be seen for the simulation data set, but not for the robustness, such as the index RR.

---

> ### Author Response · Authors · 2024-11-23
> **Response to Reviewer aCKU (1/2)**
>
> Thank you for your feedback and recognition of our work. Below are our responses to your questions and comments, and we hope they will also address your concerns:
>
> > W1: The experimental setup needs to be improved. I have my doubts about the experimental setup. A major contribution of this paper is that high outlier rates can also perform well. As for the evaluation of high outlier rate, only on the simulation data set, however, as shown in FIG. 4, the current methods can achieve good registration. This does not speak to the strengths of your approach. On the contrary, on 3DMatch dataset, with high outlier rate, it is difficult for existing methods to achieve robust registration. It is suggested to add the experimental results of Outlier rate=99% in 3DMatch dataset, which can make this paper more convincing. "Accurate registration results are obtained even at the extreme 99% outlier rate." "Is a bit of an exaggeration.
>
> - Thank you for your valuable comments on the experimental setup. We have added the experimental results of different outlier rates on the 3DMatch dataset in **Figure 2 of the supplementary material**. By conducting additional experiments on the 3DMatch, we can better demonstrate the advantages of our method in dealing with high outlier rates. We believe that these supplementary results will make the paper more convincing.
> - We also realize that the statement may be too absolute. The performance of our method under high outlier conditions is verified based on a series of experiments, which show that our method does show better robustness than the existing technology when facing challenging data. We have revised this sentence to maintain an objective description of our method. "Experimental results have shown that this method outperforms compared methods in terms of robustness and accuracy, can obtain promising registration results even in 99\% outlier rates."
>
> > W2: Some typos. What do we mean by ER and Et in Tables 3 and 4 and 5? Should it be RE and TE?
>
> - We have corrected ER and Et to RE and TE in the revised manuscript to maintain consistency with the previous definitions. At the same time, we have checked and ensured the accuracy and consistency of all terms in the manuscript.

---

> ### Author Response · Authors · 2024-11-23
> **Response to Reviewer aCKU (2/2)**
>
> > Q1: For the experiments on 3DMatch/3DLoMatch dataset to be registered with only one feature, I think it is not enough, please refer to the experimental setup of SC2-PCR and MAC. In addition, 3DMatch uses FCGF and 3DLoMatch uses Predator for feature extraction, which is kind of intentional to show the reader better results.
>
> - Referring to SC2-PCR and MAC, we have introduced two other feature descriptors to the 3DMatch and 3DLoMatch experiments. Specifically, 3DMatch uses both FCGF and FPFH. 3DLoMatch uses both FCGF and Predator (**Table 3 and Table 4 in revised manuscript**). In this way, all experimental settings are the same as KITTI, using two different feature descriptors to more comprehensively evaluate the effectiveness of the methods. We are pleased to find that even under different conditions, our method can still achieve the best results. We have detailed these experimental settings and show the performance of our method under different feature descriptors in the revised manuscript.
>
> > Q2: It is suggested that ablation experiments should also be done on the 3DMatch data set. Only the contribution to accuracy can be seen for the simulation data set, but not for the robustness, such as the index RR.
>
> - Thank you for your suggestion. We conducted ablation experiments on the Stanford dataset, and these experimental results effectively demonstrate the contribution of the components of our method in improving the registration accuracy. As for why we did not conduct ablation experiments on more datasets, the following are our considerations: Stanford is a simulated dataset, and we hope that all simulation experiments are completed on the same dataset. There is no need to consider the impact of different dataset characteristics on the experimental results, ensuring that the differences between experiments are only caused by the ablated components. In addition, the ablation experiments on Stanford have been able to verify the effectiveness of each component in our method. Even if changes are made to the dataset, we believe that our method can still maintain high performance.
>
> - It is worth mentioning that we have further improved the ablation experiments. We have added columns to Table 6 to compare knowledge sharing and convergence stages in the fixed scaling setting. As shown in Table 6, adaptive scale adjustment reduces the number of stages by about half, while effectively reducing the rotation error and translation error. Nevertheless, we recognize the value of ablation experiments on more datasets. We will consider incorporating more experiments in future work to fully evaluate the performance of our method.
>
> - In response to your question about the loss of the robustness indicator RR on the simulated dataset, we have added the content and analysis of the RR indicator (**Table 2 in revised manuscript**). These additional results further demonstrate the robustness of our method. Thank you for your thoughtful feedback.
>
> Finally, we would like to express our gratitude again for your time and effort in reviewing our paper. Please do not hesitate to let us know if you have any further concerns or comments. We would be happy to address them.

---

> > ### Comment · Reviewer_aCKU · 2024-12-03
> > **Thanks for your response.**
> >
> > You has addressed all of my concerns. Thank you for your efforts to reply and supplement the experiments. I will keep the high score unchanged.

---

> > > ### Author Response · Authors · 2024-12-03
> > > **Response to Reviewer aCKU**
> > >
> > > We sincerely appreciate the time and effort you dedicated to reviewing our work and helping us improve the paper. Thank you for your recognition of the paper and your positive comments.

---

> ### Author Response · Authors · 2024-11-26
> **Official Comment by Authors**
>
> Dear reviewer aCKU,
>
> We sincerely appreciate your time and effort in reviewing our paper and providing valuable suggestions. Special thanks for your recognition of the simple and feasible  idea and good performance of our paper.
>
> We hope to have addressed your concerns adequately. We are more than happy to answer any additional questions during the rebuttal period. Thank you again for your time and effort.
>
> Best Regards,
>
> The Authors

---

> > ### Author Response · Authors · 2024-11-30
> > **Looking forward to your feedback**
> >
> > Dear reviewer aCKU,
> >
> > We sincerely thank you again for your valuable feedback and positive reviews. As the author-reviewer discussion period is coming to an end, we would like to remind you that we have published our response to your valuable comments. Specifically:
> >
> > - We have conducted experiments with different outlier rates on 3DMatch to demonstrate the robustness of our method under high outlier rates.
> >
> > - We have modified the inappropriate expressions and unified the metric names for improved understanding.
> >
> > - We have added two additional descriptors on the 3Dmatch and 3DLoMatch datasets to provide more comprehensive experimental results.
> >
> > - We have supplemented the RR metric for the Stanford dataset to demonstrate the high registration recall of our method.
> >
> > We truly appreciate your thoughtful consideration and positive feedback. If you have any additional concerns or comments, please feel free to let us know. We would be happy to address them. Thank you again for your valuable comments and time in evaluating our work.
> >
> > Sincerely,
> >
> > Authors

---

### Author Response · Authors · 2024-12-03
**Global response**

Dear all reviewers and AC,

We would like to thank the reviewers for their insightful feedback, which significantly improved the manuscript. We have included the responses to each reviewer’s comments and highlighted the changes in our revised manuscript in blue. Thank you again for your time and significant suggestions.

During the rebuttal phase, we are pleased to receive numerous positive remarks, such as the original and novel idea (dmuo, k1rY), as well as the good performance and convincing results (aCKU, k1rY). The reviewers also appreciate the clarity and written quality of our paper (dmuo, k1rY), as well as the meaningful and potential widespread adoption (aCKU, k1rY). More importantly, inspired by the reviewers' comments and the interaction, our manuscript has been continually improved regarding the unclear parts or experimental verification about some specific points. For your reference, we supplement the main additions with the following:

- Results of 3DMatch with different outliers rates: Figure 2 in the supplementary material.
- Performance of two descriptors on the 3Dmatch and 3DLoMatch: Tables 3 and 4 in the revised manuscript.
- Robustness metric on the Stanford: Table 2 in the revised manuscript.
- Complete picture of performance: Table 1 in the revised manuscript.
- More ablation experiments: Table 6 in the revised manuscript.
- Specific definitions and detailed explanations of the roles of terms: Equations 6-10 in the revised manuscript.
- More clear visualizations: Figure 4 in the revised manuscript and Figure 1 in the supplementary material.
- Detailed introduction to the datasets: Table 1 in the supplementary material.
- Comparison of different scale-adaptive schemes: Table 2 in the supplementary material.
- More visualizations for indoor and outdoor scenes: Figures 3 and 4 in the supplementary material.

Thank you again for your thoughtful feedback and for helping us refine our work further.

Best regards,

Authors

---

### Note · Program_Chairs · 2025-01-08
**Submission Desk Rejected by Program Chairs**

The paper is desk rejected due to substantial overlap with the CVPR paper "[Adaptive Annealing for Robust Geometric Estimation](https://openaccess.thecvf.com/content/CVPR2023/papers/Sidhartha_Adaptive_Annealing_for_Robust_Geometric_Estimation_CVPR_2023_paper.pdf)" by Chitturi Sidhartha, Lalit Manam and Venu Madhav Govindu. The overlaps are: nearly identical figure 1, overlapping formulas, and nearly identical algorithm 1 box. This decision was confirmed by multiple senior members of the review committee.